# Comparison between Target Sample Check Illuminator and White Light Observation in Discriminating the Presence of Target Specimen for Endoscopic Ultrasound-Guided Fine Needle Aspiration Sample

**DOI:** 10.3390/jcm12062139

**Published:** 2023-03-09

**Authors:** Hiroki Koda, Kazuya Matsumoto, Soichiro Kawata, Yohei Takeda, Takumi Onoyama, Yuta Seki, Yuri Sakamoto, Takuya Shimosaka, Wataru Hamamoto, Taro Yamashita, Hiroki Kurumi, Naoyuki Yamaguchi, Hisashi Noma, Hajime Isomoto

**Affiliations:** 1Department of Gastroenterology, Tottori University Hospital, Yonago 683-8504, Japan; hkoda@tottori-u.ac.jp (H.K.);; 2Department of Gastroenterology and Hepatology, Nagasaki University Graduate School of Biological Sciences, Nagasaki 852-8501, Japan; 3Department of Data Science, The Institute of Statistical Mathematics, Tachikawa 190-8562, Japan

**Keywords:** target sample check illuminator, endoscopic ultrasound-guided fine needle aspiration, endoscopic ultrasound-guided fine needle biopsy, rapid on-site evaluation, macroscopic on-site quality evaluation

## Abstract

Endoscopic ultrasound-guided fine needle aspiration (EUS-FNA) is an essential endoscopic tissue sampling method for diagnosing pancreatobiliary diseases; however, determining the presence of target specimens mixed in the blood by conventional observation is challenging due to the small size of the obtained sample. This study investigated the usefulness of a target sample check illuminator (TSCI) that emits a specific wavelength of light to determine the presence of target specimens. Twenty-seven patients who underwent EUS-FNA at our hospital were included. Conventional white light observation was performed for the collected samples, followed by TSCI; six people evaluated the presence of the target specimen on a 5-point scale. The target specimen discrimination score using TSCI (median: 5) was significantly higher than that using conventional white light observation (median: 1) (*p* < 0.001). No significant difference was observed in the discrimination score between the evaluator (novice vs. expert, *p* = 0.162) and puncture needle (22G vs. 25G, *p* = 0.196). The discriminability of TSCI in the samples obtained using EUS-FNA was significantly higher than that of conventional observation. TSCI does not depend on the evaluator or puncture needle for the identification of the target specimen; hence, it can provide a good pathological specimen and may contribute to the improvement of the diagnostic ability.

## 1. Introduction

The tissue samples of target lesions for pancreatobiliary diseases were difficult to collect, especially for those patients with pancreatic diseases; hence, treatment decisions were often made based on imaging diagnoses without histological examinations. As a result, a certain number of patients could not be surgically treated or have chemotherapy, radiation therapy, or follow-up care in accordance with the true diagnosis, and, thus, could not be provided with appropriate medical care [1]. Therefore, in 1980, Strohm et al. reported the usefulness of endoscopic ultrasonography (EUS) for visualizing adjacent organs through the gastrointestinal tract, which was an epoch-making event in the clinical practice of pancreatobiliary diseases [2]. Furthermore, in 1992, Vilmann et al. reported endoscopic ultrasound-guided fine needle aspiration (EUS-FNA), which is a technique that allows for tissue sampling under EUS and pathological evaluation for pancreatic lesions that were difficult in the past [3].

Recently, EUS, with its high image resolution, plays a central role in the diagnosis of pancreatobiliary diseases, subepithelial lesions (SELs), and enlarged lymph nodes. Since the differentiation between benign and malignant diseases significantly impacts the treatment plan, it is especially important to perform a definitive pathological diagnosis; EUS-FNA is widely used as a tissue sampling method [4,5,6]. However, the sampled tissue is small and mixed with blood, so it is often difficult to determine whether the appropriate target specimen has been acquired with conventional observation. Rapid on-site evaluation (ROSE) [7,8], which involves immediate cytological diagnosis, has been reported as a solution to this problem. However, the shortage of cytologists in Japan is a chronic problem, and this causes the introduction of ROSE to be difficult. Therefore, to improve the diagnostic ability, it is necessary to increase the number of punctures to ensure a sufficient sample volume; however, this may increase the risk of adverse events and peritoneal dissemination [9,10]. To address these problems, it is important to ensure it is easy for novices to evaluate the samples. We have developed a target sample check illuminator (TSCI) to determine the presence of target specimens in the samples acquired using EUS-FNA without the loss of quantity and quality and have reported its usefulness for pancreatic tumors, swollen lymph nodes, and SELs [11,12]. However, the versatility of TSCI has not been verified since evaluations using TSCI are performed with special wavelengths of light not usually experienced in medical practice.

This study aimed to examine the usefulness of TSCI for EUS-FNA samples, the experience of using TSCI, and differences in the ability to identify target specimens with the different puncture needle diameters used for EUS-FNA.

## 2. Materials and Methods

### 2.1. Study Design and Population

This was a single-center, retrospective, cohort study conducted at Tottori University Hospital in accordance with the ethical standards set forth in the Declaration of Helsinki in 1964; it was approved by the Institutional Review Board of our University Hospital (approval number: 2335). Twenty-seven patients who underwent EUS-FNA and presented with lesions detected using abdominal ultrasonography, computed tomography, magnetic resonance imaging, and esophagogastroduodenoscopy that required pathological diagnosis at our hospital from June to September 2014 were enrolled (Table 1).

### 2.2. EUS-FNA Procedure

We used a GF-UCT260 endoscope (Olympus Optical, Tokyo, Japan) and an EU-ME2 processor (Olympus Optical, Tokyo, Japan) in all cases. The puncture needle was selected from 19G, 22G, and 25G (Expect™, Boston Scientific. Marlboro, MA, USA) by an endoscopist, based on the patient background and lesion characteristics. All EUS-FNA procedures were performed through the esophagus, stomach, and duodenum, under midazolam conscious sedation. After visualizing the lesion and confirming the lack of vessels in the puncture pathway using EUS, the FNA needle was punctured to collect tissue samples with approximately 20 back-and-forth movements using 20 mL of suction. We performed both the conventional and TSCI methods to determine whether the target specimen was obtained. The procedure was terminated if a sufficient amount of sample was successfully collected; otherwise, additional punctures were performed when the sample was insufficient.

### 2.3. TSCI (Target Sample Check Illuminator)

Based on the light absorption spectrum of human oxyhemoglobin (peak spectrum of deoxidized hemoglobin/oxidized hemoglobin: 550 nm/540 nm and 585 nm) reported by Zijlstra et al. in 1991 [13], we found that blood components and tissues could be clearly distinguished by transmitting a wavelength of 605 nm light through the collected samples [11]. Observing the specimens with transmitted light of this wavelength, blood absorbs light and appears dark brown, whereas tissue appears orange without light absorption, causing it to be easier to distinguish tissue components in specimens with mixed blood. TSCI is a device developed by Adachi Co., Ltd. (Osaka, Japan) that facilitates the identification of the presence of target specimens collected using EUS-FNA using transmitting 605 nm wavelength light from below. Currently, we use TSCI for evaluating the EUS-FNA samples in routine clinical practice at our hospital (Figure 1).

### 2.4. Evaluation of Sampling Tissue

The EUS-FNA samples were extruded from the puncture needle into a plastic Petri dish, and we evaluated whether the target specimen had been acquired by observation using the conventional method with white light. The presence of a target specimen is usually determined by the appearance of a white component in the thread-like red specimen (blood component). Subsequently, TSCI was performed. The Petri dish was set in the TSCI and irradiated with a 605 nm wavelength light to observe the transmitted light. Unlike conventional observation with white light, blood components appear dark brown and target specimens appear orange; therefore, the presence of the target specimen is determined based on these findings (Figure 2). The images of EUS-FNA specimens for discriminant scoring were taken at a fixed distance from directly above the specimen for both white light and TSCI observations. Therefore, there is no difference in the image quality from case to case. The solid component was fixed in formalin for histological diagnosis, and the liquid component was pathologically diagnosed using cytological diagnosis.

### 2.5. Study Protocol, Outcome Measures, and Statistical Analysis

Using the images of the EUS-FNA samples with conventional and TSCI observations, six evaluators rated whether the target specimens were visible in the samples mixed with blood on a five-point scale (1: bad, 2: poor, 3: moderate, 4: good, and 5: excellent). White light conventional observation images were rated first, followed by TSCI images, and the evaluators were blinded to discriminant scoring among themselves. The evaluators consisted of three experienced TSCI users and three TSCI novices. The TSCI novices were provided with a 15 min explanation of the TSCI observation methods with examples before sample evaluation. The statistical analyses were performed using SPSS ver. 25 (IBM, Armonk, NY, USA). The stratified categories for each clinical parameter were evaluated using the Wilcoxon signed-rank test, Paired Wilcoxon signed-rank test, and Chi-square tests. The Wilcoxon signed-rank test was used for the variables that were not normally distributed. The statistical significance was set at *p* < 0.05.

## 3. Results

The median age of the patients was 65.3 years (range: 23–80), with 17 men and 10 women. EUS-FNA was performed for 18 pancreatic tumors, three gastric SELs, three lymph nodes, and two others with three puncture needle sizes: 22G (13 cases, 48.1%), 25G (10 cases, 37%), and 19G (4 cases, 14.8%). As for the final diagnosis, with 10 cases, pancreatic cancer was the most commonly observed (37%), followed by 3 autoimmune pancreatitis cases (11.1%), 3 gastrointestinal stromal tumors (11.1%), 2 solid pseudopapillary neoplasms (7.4%), 2 intrapancreatic accessory spleen (7.4%), 2 benign reactive enlarged lymph nodes (7.4%), 1 lymphoepithelial cyst (3.7%), 1 liposarcoma (3.7%), 1 diffuse large B-cell lymphoma (3.7%), 1 papillitis of Vater (3.7%), and 1 esophageal achalasia (3.7%). No procedure-related adverse events were observed.

The diagnostic ability of EUS-FNA was as follows: sampling success rate, 96.3% (26/27); sensitivity, 88.2% (15/17); specificity, 90% (9/10); positive predictive value, 93.8% (15/16); negative predictive value, 81.8% (9/11); and accuracy, 88.9% (24/27) (Table 2). The discrimination scores for the target specimens using conventional observation were 1: 65.4%, 2: 13.6%, 3: 10.5%, 4: 7.4%, and 5: 3.1% (median 1, interquartile range [IQR] 1–2), while those using TSCI were 1: 7.5%, 2: 2.4%, 3: 9.3%, 4: 30.2%, and 5: 50.6% (median 5, IQR 4-5), resulting in significantly higher discrimination scores in the TSCI group (*p* < 0.001) (Figure 3). The TSCI group had a high discrimination ability, scoring 4 or higher, compared to the conventional observation group (88.2% vs. 5.2%, *p* < 0.001). In the TSCI observation group, there was no significant difference in the number of cases with a score of 4 or higher between novices and experienced raters (76.5% vs. 85.2%, *p* = 0.162), indicating that the experience with TSCI did not the affect discriminatory ability (Figure 4). Moreover, there was no significant difference in the discrimination ability between 22G and 25G puncture needles (median 5; IQR4–5 vs. 4; IQR4–5, *p* = 0.196) (Figure 2 and Figure 5).

## 4. Discussion

Pancreatic and biliary tract cancers have very poor prognoses; therefore, it is important to detect and treat them as early as possible [14]. EUS-FNA is a fundamental diagnostic technique for pancreatobiliary diseases and is useful for many other lesions, such as gastrointestinal SELs, lymph nodes, and intraperitoneal masses in a wide area from the mediastinum to the pelvis. Therefore, EUS-FNA is required to safely collect appropriate samples with a minimum number of punctures, and many studies have attempted to improve the diagnostic ability, increase the number of specimens, and reduce adverse events [15,16,17,18]. Specific items to improve diagnostic performance include a selection of the EUS scope, needle shape, needle diameter, number of punctures, puncture method, number of strokes, aspiration method, and specimen processing method. Regarding scope selection, it was reported that forward and anterior oblique viewing scopes had comparable results concerning both the diagnostic ability and safety, as well as the shape and puncture needle diameter. The use of a large-diameter (19G > 22G) or a Francine needle had a better diagnostic ability, and the adverse events rate was comparable [19,20]. It has also been reported that the diagnostic ability can be increased with lesser punctures by using the door-knocking method for a quick puncture and the fanning method, wherein the needle is repeatedly punctured while shifting its position within the target lesion [16,21]. Furthermore, when the inner wire is slowly pulled out while the puncture is repeated, it applies only a small amount of negative pressure to the needle lumen, resulting in less blood in the specimen, thereby increasing the diagnostic sensitivity [22]. However, it is also important to reduce the number of adverse events parallel to these attempts to improve the diagnostic ability of EUS-FNA. EUS-FNA is considered a safe tissue sampling method with an adverse events rate of <1% [23], and the frequency of the occurrence of bleeding and pancreatitis is at 0.23% and 0.28% [24,25], respectively, but it should be noted that the incidence can sometimes be severe [26]. In particular, needle tract seeding, which is the formation of seeding nests along the puncture path after EUS-FNA for malignant disease, has been reported to occur in 0.33% of cases, which can have a significant impact on the treatment plan and the prognosis of patients and is one of the topics of a recent study [27]. To reduce these adverse events, we believe that it is desirable to minimize the needle diameter and the number of puncture attempts.

TSCI is a tissue discrimination instrument that uses a specific wavelength of light of 605 nm to facilitate the determination of the presence of target specimens in EUS-FNA samples. The use of this device is expected to improve the pathological diagnostic ability by ensuring the sample quality and reduce the adverse events by preventing unnecessary punctures. In 2021, TSCI was significantly more useful in discriminating target specimens with 95.8% for TSCI observation than normal observation with 75.0% (*p* = 0.025) [12]. However, the degree of improvement in visibility was unknown because the evaluation was based on two choices: visible or invisible. In our study, the degree to which the target specimen could be discriminated was scored on a five-point scale, albeit subjectively, and the median discrimination score was 1 (IQR: 1–2) for normal observation and 5 (IQR: 4–5) for TSCI observation. The use of TSCI resulted in a large score increase, proving the superiority of TSCI by a significant margin over its usefulness in the previous report (*p* < 0.001). 

One important aspect of this study was that it included 37% of cases in which EUS-FNA was performed using a 25G puncture needle, which has not been examined in previous reports. In this study, the discrimination ability of the 25G needle was not significantly different from that of the 22G needle, suggesting that TSCI is useful even for thin samples collected using a 25G small-diameter needle. Although we sometimes use a large-diameter needle for EUS-FNA to secure sample volume, a small-diameter needle is often used in the case of difficulty in a puncture or operable lesion in which the risk of seeding needs to be minimized [28]. In such a situation, the contribution of TSCI will be even greater because sample evaluation using conventional observation is difficult due to the small sample size. It is also important to note that the TSCI did not consume the collected samples. The usefulness of ROSE as a method to improve the diagnostic ability of EUS-FNA has been widely reported [7,8]. This can eliminate unnecessary punctures, as a cytological evaluation of the sample is immediately performed at the time of the EUS-FNA, while consuming a sample for cytology reduces tissue for histology and may cause a histological evaluation, such as structural atypia, more difficult. Furthermore, as precision medicine based on genetic information is expected to develop further in the future, securing the number of target specimens using EUS-FNA will become increasingly important [29]. TSCI is also expected to be a significant contribution in this regard. 

Another important verification in this study is whether sample evaluation using TSCI is possible for everyone. ROSE is diagnosed by cytologists or pathologists trained according to the evaluation criteria established for cytology; therefore, not everyone can perform ROSE. TSCI is required so that everyone can evaluate the presence of target specimens at the same level. In this study, there was no significant difference between novices and experienced raters in cases where the discrimination score of the target sample was four or higher. In other words, with TSCI observation, an equal discriminative ability can be expected for all, if a brief and simple explanation is provided. The simplicity of use and the fact that novices can discriminate the target specimens suggest that this device will be useful for medical institutions in Japan, where there is a shortage of cytologists, so we look forward to its widespread use. 

Although we have discussed the usefulness of TSCI, it is necessary to understand its disadvantages. TSCI is effective only in identifying the target specimen, not in differentiating between benign and malignant tumors. In particular, pancreatic cancer, one of the most common carcinomas for which EUS-FNA is performed, is characterized by abundant stroma and fibrosis in the surrounding tissues. Even if a sample containing a large amount of target specimen is obtained, the actual number of cancer cells may be negligible, and no definitive diagnosis may be reached. However, the macroscopic on-site quality evaluation (MOSE) study reported by Iwashita et al. showed that the sensitivity of malignancy was significantly higher when the grossly visible tissue component in the sample obtained using EUS-FNA was 4 mm or longer than when it was 4 mm or shorter (95.5% vs. 57.1%, *p* < 0.0001) [18]. This finding that a longer gross tissue component improves the malignant diagnostic ability may also be applicable for diagnosis using TSCI. Nevertheless, this point needs further verification based on a pathological diagnosis. Therefore, the TSCI used in our study was a device that contributed to bridging the gap between ROSE and MOSE by reducing the number of EUS-FNA-related adverse events and solving the disadvantages of ROSE, the consumption of specimens, and MOSE, the inability to identify the target specimens contained in a small thin sample.

A limitation of this study is that it was a single-center retrospective study with small sample size. In addition, the discriminative ability of the target specimen was scored subjectively, causing it to be difficult to perform a constant objective evaluation. Moreover, the TSCI images were evaluated immediately after the conventional observation images, which may result in a bias. To further validate its usefulness, it is desirable to create an objective evaluation index in addition to prospectively accumulating the number of cases at multiple centers. 

In conclusion, the discrimination ability of the target specimens in samples obtained using EUS-FNA was significantly higher with TSCI observation than with conventional white light observation, regardless of the puncture needle diameter. There was no difference in the discrimination ability between experienced and novice raters, indicating that specimen evaluation using TSCI can be performed for samples of various sizes by any evaluator.

## Figures and Tables

**Figure 1 jcm-12-02139-f001:**
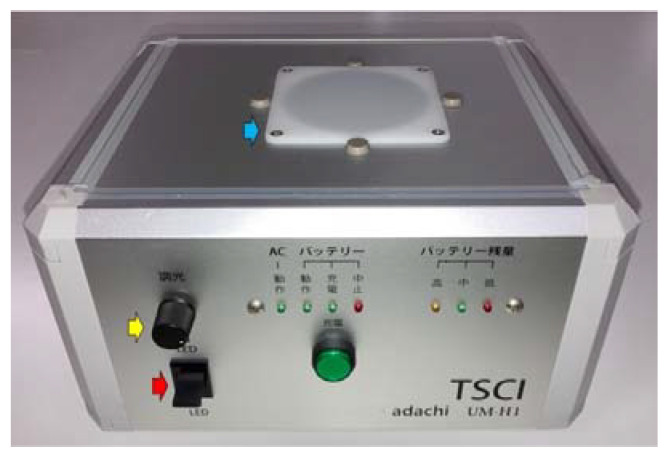
Main unit of TSCI. On the front of the device, there is a tab to switch the 605 nm wavelength light (red arrow), a knob to adjust the light intensity (yellow arrow), and a lamp to check the operation status. In the center of the upper surface of the device, there is a white plate that emits transmitted light from below, and a Petri dish containing the specimen is placed on this plate (blue arrow). Abbreviation: TSCI, target sample check illuminator.

**Figure 2 jcm-12-02139-f002:**
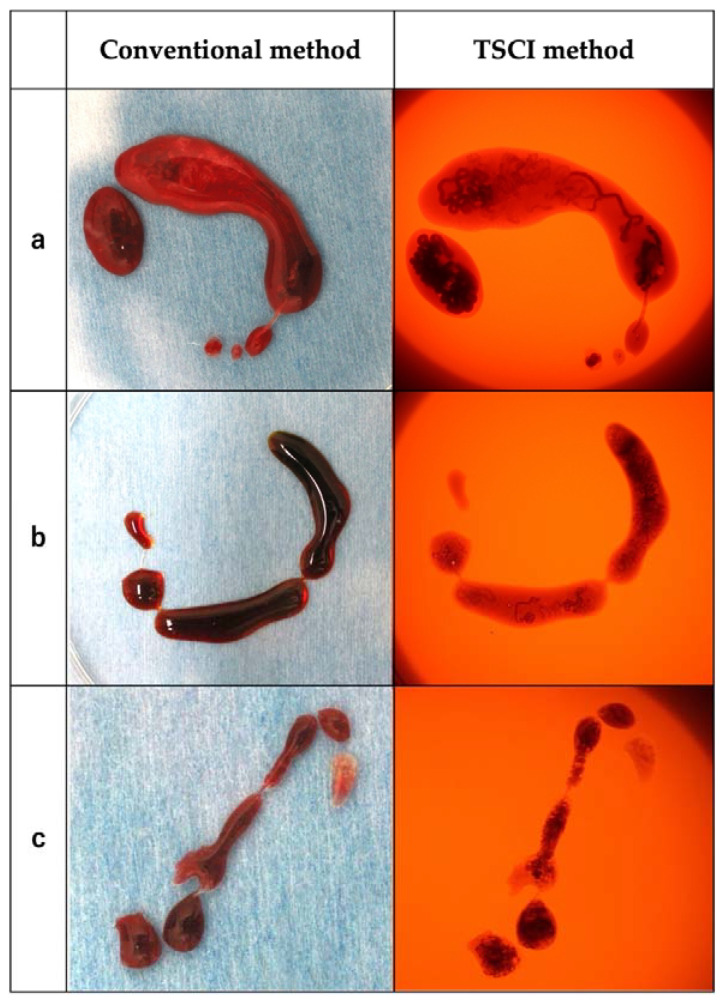
Representative EUS-FNA sample images. The images of the EUS-FNA samples from the three cases with puncture needles of different sizes were observed using the conventional and TSCI methods. The solid components and blood and tissue components are clearly distinguished by the transmitted light in the TSCI method. (**a**) AIP, needle size: 19G, median score (conventional/TSCI): 3.5/5; (**b**) Pancreatic cancer, needle size: 22G, median score (conventional/TSCI): 1/5; and (**c**) Pancreatic cancer, needle size: 25G, median score (conventional/TSCI): 3.5/5. Abbreviations: AIP, autoimmune pancreatitis.

**Figure 3 jcm-12-02139-f003:**
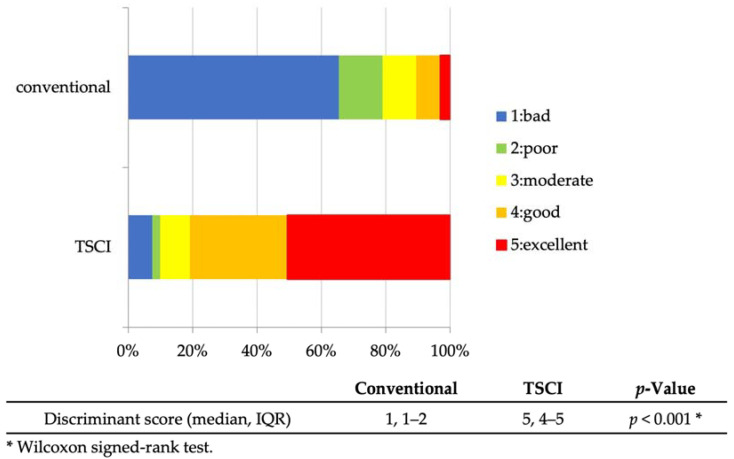
Discriminant scores for EUS-FNA sample (conventional vs. TSCI). Discriminant scores for EUS-FNA samples by the six raters. The TSCI observations had significantly higher discriminant scores than the conventional method. Abbreviations: EUS-FNA, endoscopic ultrasound-guided fine-needle aspiration; TSCI, target sample check illuminator; and IQR, interquartile range.

**Figure 4 jcm-12-02139-f004:**
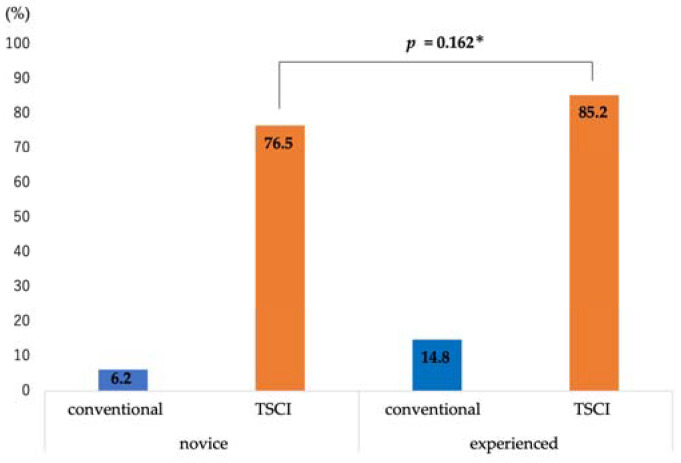
Differences from experience in high discriminant scoring cases. In high-scoring cases (≥4), there was no significant difference between the novice and experienced raters. * Chi-square test. Abbreviations: TSCI, target sample check illuminator.

**Figure 5 jcm-12-02139-f005:**
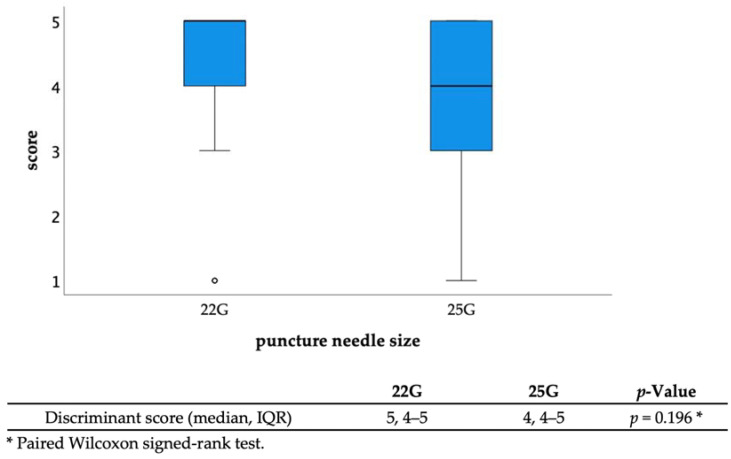
Discriminant scores with different needle sizes (22G vs. 25G). There was no significant difference in the discriminant scores between different puncture needle sizes. Abbreviations: IQR, interquartile range.

**Table 1 jcm-12-02139-t001:** Patient characteristics.

Patient Characteristics	*N* = 27
Age, median (range)	71.5 (38–85)
Male:Female, *n*	17:10
Puncture site and final diagnosis	
Pancreatic tumor, *n* (%)	18 (66.7)
PC, *n* (%)	10 (37.0)
AIP, *n* (%)	3 (11.1)
SPN, *n* (%)	2 (7.4)
Accessory spleen, *n* (%)	2 (7.4)
LEC, *n* (%)	1 (3.7)
Gastric subepithelial lesion, *n* (%)	3 (11.1)
GIST, *n* (%)	3 (11.1)
Lymph node, *n* (%)	3 (11.1)
reactive swelling, *n* (%)	2 (7.4)
DLBCL, *n* (%)	1 (3.7)
Retroperitoneal tumor, *n* (%)	1 (3.7)
Liposarcoma, *n* (%)	1 (3.7)
Ampulla of Vater, *n* (%)	1 (3.7)
Inflammatory change, *n* (%)	1 (3.7)
Others, *n* (%)	1 (3.7)
Esophageal achalasia, *n* (%)	1 (3.7)
Diameter of lesions (mm), median (range)	25.5 (7–51)
Puncture needle, *n* (%)	
19G/22G/25G	4 (14.8)/13 (48.1)/10 (37)

Abbreviations: PC, pancreatic cancer; AIP, autoimmune pancreatitis; SPN, solid pseudopapillary neoplasm; LEC, lymphoepithelial cyst; GIST, gastrointestinal stromal tumor; and DLBCL, diffuse large B-cell lymphoma.

**Table 2 jcm-12-02139-t002:** Diagnostic ability of EUS-FNA.

Parameter	Diagnostic Ability
Sampling success rate	96.3% (26/27)
Sensitivity	88.2% (15/17)
Specificity	90% (9/10)
PPV	93.8% (15/16)
NPV	81.8% (9/11)
Accuracy	88.9% (24/27)

Abbreviations: PPV, positive predictive value; and NPV, negative predictive value.

## Data Availability

The data presented in this study are available on request from the corresponding author. The data are not publicly available due to privacy or ethical.

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
