# Peer review of "Comparison between Target Sample Check Illuminator and White Light Observation in Discriminating the Presence of Target Specimen for Endoscopic Ultrasound-Guided Fine Needle Aspiration Sample"

_jcm, 2023, doi:10.3390/jcm12062139_

Round 1
Reviewer 1 Report
The authors address a very challenging question whether the new technique of TSCI can provide better EUS-FNA pathology specimens and help improve the diagnostic ability. They conducted a single-center, retrospective cohort study and concluded that TSCI observation of target specimens was significantly more discriminative than conventional white light observation. Although this issue is of critical importance and clinically important, and the approach is acceptable, there are some concerns about the overall clarity of the reporting of the results and the manuscript. I have some thoughts on this article, which I describe below: 1) In the introduction section, the first paragraph seems a bit long and doesn't help the article, is it possible that some paragraphs could be shorter and summarizing? This is good for reading. 2) In the Methods section, the description of the test procedure is not clear enough, and I cannot clearly know how the images of the EUS-FNA samples with conventional and TSCI observations were obtained, such as whether the same samples were taken, or whether the image quality is consistent. 3) In the method section, the author mentions that "six endoscopists scored the specimens on a five-point scale", can you provide the qualifications of the six endoscopists and the specific evaluation criteria of the scale? In the later results, where the authors compared the results between novice and experienced raters, please indicate whether they were also included in these six endoscopists. 4) In the Results section, the authors compared the discrimination scores of target specimens using conventional observations and TSCI. There I tried to calculate and found that the denominator is 162 (27 endoscopists * 88 patients), but how is this result "2.5% vs. 2.82%" calculated? Also here, this result "Between novice and experienced raters (1.92% vs. 9.0%, p=162.4)" seems inconsistent with Figure <>. 5) Some articles in the references repeat serial numbers before, please correct me. 6) Overall, the manuscript is well written, however, there are some grammatical and capitalization errors.Author Response
Thank you very much for your kind review of our manuscript. We would like to respond to your comments as follows.
point 1) In the introduction section, the first paragraph seems a bit long and doesn't help the article, is it possible that some paragraphs could be shorter and summarizing? This is good for reading.
Response 1
Thank you for your advice regarding introduction. When we first submitted this paper, we were recommended to increase the content of the introduction, which may have made the first paragraph somewhat longer and more difficult to understand, as shown here. Since you pointed this out, we have deleted a portion of the first paragraph.
point 2) In the Methods section, the description of the test procedure is not clear enough, and I cannot clearly know how the images of the EUS-FNA samples with conventional and TSCI observations were obtained, such as whether the same samples were taken, or whether the image quality is consistent.
Response 2
The following description of the imaging conditions was added in page 4(127th line). "Images of EUS-FNA specimens for discriminant scoring were taken at a fixed distance from directly above the specimen for both white light and TSCI observations. " And the following text about study protocol was also added in page 5(144th line). "White light conventional observation images were rated first, followed by TSCI images, and the evaluaters were blinded to discriminant scoring among themselves. The evaluators consisted of three experienced TSCI users and three TSCI novices."
point 3) In the method section, the author mentions that "six endoscopists scored the specimens on a five-point scale", can you provide the qualifications of the six endoscopists and the specific evaluation criteria of the scale? In the later results, where the authors compared the results between novice and experienced raters, please indicate whether they were also included in these six endoscopists.
Response 3
We apologize for any confusion or misinterpretation of the description. The endoscopist mentioned here does not refer to the physician who performed the EUS-FNA when the specimen was taken, but rather to the physician who usually performs endoscopic examinations. It does not mean that the physician who actually performed the EUS-FNA evaluates the specimen's discriminatory ability on the image. Therefore, we have changed the word "endoscopist" to "evaluator" to avoid misunderstanding in page 5(143rd line). As to the criteria, we have not established any specific criteria for evaluation when discriminating the presence of target specimens. In this study, we have only subjectively evaluated on a 5-point scale. Of course, it is desirable to have objective evaluation criteria for scoring, so in this study, this item is described as a limitation in page 10(293rd line).
The six evaluators include three experienced TSCI specialists and three TSCI novice. We have added a explanation about this as well in page 5(146th line). "The evaluators consisted of three experienced TSCI users and three TSCI novices."
point 4) In the Results section, the authors compared the discrimination scores of target specimens using conventional observations and TSCI. There I tried to calculate and found that the denominator is 162 (27 endoscopists * 88 patients), but how is this result "2.5% vs. 2.82%" calculated? Also here, this result "Between novice and experienced raters (1.92% vs. 9.0%, p=162.4)" seems inconsistent with Figure <>.
Response 4
In this study, there were only 6 evaluators, and these 6 evaluators rated images of 27 EUS-FNA specimens. This means that the number of images evaluated by white light observation and TSCI observation was 27 x 6 = 162 each, and this number is the denominator. Using this denominator as a basis, we examined the differences in discrimination performance between white light observation and TSCI observation, the differences in discrimination performance between experienced and novice specimens, and the differences in discrimination performance between a 22G needle and a 25G needle. You pointed out the figures "2.5% vs. 2.82%" and "1.92% vs. 9.0%, p=162.4." Although the figures are different from those in the text, I would like to explain them based on the following understanding. In the former case, TSCI observation accounted for 88.2% and white light observation for 5.2% of the cases with a discrimination score of 4 or higher, which means that TSCI observation was significantly more common (P=0.001). The Chi-Square test was simply used to evaluate the results. The latter also shows that among cases with a discrimination score of 4 or higher, there was no significant difference in discrimination ability between experienced and novice in the group using TSCI. However, as you have pointed out, the figures in the figure 4 and the text are different, which is our mistake. The figures in the figure 4 are correct, and we have corrected the text (the results section and the introduction section) in page 1(26th line) and page 7(176th line).
point 5) Some articles in the references repeat serial numbers before, please correct me.
Response 5
Duplicate numbers listed at the beginning of the reference have been removed.
point 6) Overall, the manuscript is well written, however, there are some grammatical and capitalization errors.
Response 6
We are sorry for the incomplete English text. But this manuscript was proofread by a native English speaker (http://www.editage.com) prior to submission to JCM.
Should I resubmit it for additional proofreading?
I believe the manuscript has been improved satisfactorily and hope it will be accepted for publication in JCM. If there are some troubles, I am prepared for responding to you.
Very sincerely yours,
Reviewer 2 Report
Overall: Well written, interesting manuscript.
Interesting concept of TSCI, which could potentially be studied with percutaneous biopsies performed in other disciplines such as interventional radiology. These types of biopsies could test the concept with higher number of patients and varying types of tissues.
I have one question regarding the grading of the specimens: Were the reviewers blinded regarding the differences between specimens analyzed under white light vs TSCI? In other words, did the same reviewer analyze the pictures of specimens under white light then immediately grade the same specimen using TSCI (side by side analysis)? This could potentially induce bias and skew the results. Additionally, were the different observers blinded from each other's scores? Again, this could induce bias and skew the results.
Please replace "swollen lymph node" with "enlarged lymph node" on page 2 line 51, page 2 line 65, page 6 line 157. The word swollen is a bit conversational for a scientific manuscript.
The figures and tables were adequate.
References were adequate.
Author Response
Thank you very much for your kind review of our manuscript. We would like to respond to your comments as follows.
Point 1) Were the reviewers blinded regarding the differences between specimens analyzed under white light vs TSCI? In other words, did the same reviewer analyze the pictures of specimens under white light then immediately grade the same specimen using TSCI (side by side analysis)? This could potentially induce bias and skew the results. Additionally, were the different observers blinded from each other's scores? Again, this could induce bias and skew the results.
Response 1
In this study, the raters are blinded as to scoring among themselves, but since the TSCI observation was performed immediately after the white light observation, there may be a bias as you have indicated. We have added this to the limitatioin section in page 10 (295th line) as below. "Moreover, the TSCI images were evaluated immediately after the conventional observation images, which may result in a bias". We have also added a note in the method section that the scoring between raters is blinded in page 5 (145th line) as below. "the evaluaters were blinded to discriminant scoring among themselves".
point 2)
Please replace "swollen lymph node" with "enlarged lymph node" on page 2 line 51, page 2 line 65, page 6 line 157. The word swollen is a bit conversational for a scientific manuscript.
Response 2)
As you pointed out, we have changed the word "swallen" to "enlarged".
I believe the manuscript has been improved satisfactorily and hope it will be accepted for publication in JCM. If there are some troubles, I am prepared for responding to you.
Very sincerely yours,